# Exploring Human–Wildlife Conflict and Implications for Food Self-Sufficiency in Bhutan

Sangay Wangchuk [1,2,*], Jennifer Bond [1], Rik Thwaites [1] and Max Finlayson [1]

1   Gulbali Institute, School of Agricultural, Environmental and Veterinary Sciences, Charles Sturt University, Albury-Wodonga, NSW 2640, Australia
2   Ugyen Wangchuck Institute for Conservation and Environment Research, Lamai Goempa 32001, Bhutan
*   Correspondence: swangchuk@csu.edu.au

**Abstract:** The conflict between humans and wildlife is a global issue in the increasingly shared landscape. Human–Wildlife Conflict (HWC) is often viewed as a threat to most of the rural populace of the world, as crop losses to wildlife remove the household's food supply, and are an economic drain on the homestead. In this paper, we study the extent of crop damage by wild animals in two districts of Bhutan: Trashiyangtse and Tsirang. We surveyed 431 respondents from the two districts and interviewed 40 central and local government officials and residents. The vast majority of respondents from both study districts (Trashiyangtse = 98.7%; Tsirang = 92.2%) reported having experienced conflicts with wild animals from 2017 to 2019. On average, respondents' households lost over half a month to more than a month's worth of household food requirements, with some households claiming to have lost over six months' worth of household food requirements, annually to wild animals. The loss of crops to wild animals removes households' food supply and discourages farming, resulting in increased fallow lands. The fallow lands which are close to human settlements, then become habitats for wild animals, aggravating the incidence of HWC, and as such are directly linked to reduced food production.

**Keywords:** crop damage; wild animals; farmland abandonment; migration; wild pig

## 1. Introduction

Human–wildlife conflict (HWC) is a global problem, and with increasingly shared landscapes between the two, HWC will continue [1]. Additional factors such as human population growth, increased pressure on land and natural resources, and climate change have also contributed to the increase in HWC worldwide [2–4]. Some examples of HWC include livestock predation, crop raids, destruction of stored grains, physical and psychological harm or death of humans and wildlife, damage to infrastructure [5–8], and disease transmission to humans and livestock [2,9].

Though the HWC literature is skewed towards impacts on human livelihoods, the literature also refers to HWC as harmful to wild animals [10,11]. For example, Dickman [12] asserted that HWC is one of the most critical threats facing many wildlife species today, largely due to killing of wild animals, either in retaliation or for the perceived risk to humans [13]. Nevertheless, HWC is often viewed as a threat to most of the world's rural populace, primarily due to crop damage, particularly in Asia and Africa [5,6,14].

In this article, we consider HWC as any interactions between humans and wildlife that negatively impact the farming community. Thus, those wildlife species that collide with some activity of human interest, from the human point of view, are classified as problematic wild animals [15]. We acknowledge that humans often overstate wildlife damage [16], and therefore we do not deny that the views expressed by our respondents regarding their interactions with wildlife are restricted to this biased perspective. However, it is important for conservation management that these perceptions are known, as there may be implications for wildlife, such as retaliation.

Worldwide, subsistence farmers have expressed their concerns about food shortages because of raiding by wildlife [6]. Since subsistence homesteads depend mainly on crops they grow for their daily nutrition, a reduction in the food supply could threaten the food sufficiency of homesteads [17] (i.e., the ability to meet consumption needs from their own production [18]). Furthermore, Quandt [19] reported that crop raiding increases dependency on purchased items, and results in substantial financial loss, particularly for poor communities and those dependent on associated industries [5,20]. In addition, some studies identify HWC as one of the important factors discouraging agriculture production and leading to an increase in fallow lands in developing countries [21–27], threatening food self-sufficiency at both household [24] and national levels [14]. Though several factors such as climate change [28,29], land fragmentation [30,31], topography, and soil fertility [32] influence a household's decision to abandon or reduce their cultivated land area, HWC is one of the reasons for increased fallow lands [22,23]. For example, Yan et al. [14] found that HWC became too much for some farming families in China to continue farming, resulting in farmland abandonment and family migration [23,33].

The migration of household members creates farm labor bottlenecks [34], resulting in a negative feedback loop in which family migration leads to a labor shortage, and consequently to an increase in fallow lands [35]. While farmland abandonment, or fallow farmlands, may increase forest cover and support ecosystem restoration [36,37], the existing literature identifies negative implications for the farming community. For example, fallow farmlands bring wild animals closer to settlement areas, leading to further intensification of HWC [22,23]. Therefore, HWC incidents can reduce household food production and impact household food self-sufficiency, resulting in increased fallow lands and dependency on purchased items [38].

Food self-sufficiency refers to the ability to meet consumption needs from one's own production (at household, region, or country level) [18]. Food self-sufficiency differs from the concept of food security, which refers to "a situation that exists when all people, at all times, have physical, social and economic access to sufficient, safe and nutritious food that meets their dietary needs and food preferences for an active and healthy life" [39] (p. 1). However, subsistence farmers largely depend on the foods they grow, because of low purchasing power [40], implying the need to understand household food self-sufficiency. Therefore, this paper focuses on the concept of food sufficiency at the household level. In the next section, we outline the context of HWC in Bhutan.

*Human–Wildlife Conflicts in Bhutan*

Bhutan's economy remains largely agrarian, despite limited agriculturally productive land. A labor force survey reported that 49.9% of the total employed Bhutanese population was engaged in the agriculture sector in 2020 [41]. However, land use and land cover surveys show that only 2.75% of the country's total area was under cultivated agriculture in 2016 [42], due to key geographical features such as topography. Despite the topographical challenges, an aspiration to be self-sufficient in food in Bhutan was called for as early as its fifth five-year plan (1981–1986) [43]. However, the country still imports most of its food, as reflected in the statistical yearbook of Bhutan, published by the National Statistical Bureau of Bhutan (NSB) [44].

Bhutanese farmers generally practice subsistence agricultural farming, with small land holdings [45], through the tradition of household labor exchange [46]. In recent years, limited agricultural production has been challenged by farm labor shortages, resulting from rural to urban migration [8,47], and climate change, through changes in monsoon patterns [48,49]. In addition to these challenges, crop damage by wild animals is a major challenge to Bhutanese farmers because of the strong nature conservation policy, which prohibits hunting wild animals [50], and the acute dependence of rural households on farming [51].

Livestock losses and crop damage by wild animals are the most common types of HWC in Bhutan. However, the National Plant Protection Centre and World Wildlife Fund

Bhutan [52] reported that crop losses are far greater in scope and magnitude than livestock loss, portraying the significance of crop damage to food production. Similarly, a survey by the Japan International Cooperation Agency (JICA) [53], reported that about 30% of crops are lost to wild animals in Bhutan. This is significant, as subsistence farmers grow crops for direct home consumption. Yeshey et al. [54] assert that losing crops to wildlife reduces food production, which translates to increased vulnerability of households' food self-sufficiency.

In addition to crop damage by wildlife, one of the implications of HWC is an abandonment of farmlands. The Renewable Natural Resources (RNR) [55] census of Bhutan reported that, for 24.6% of its respondents (N = 9368), HWC was the main reason for leaving their irrigated farmlands fallow. Recognizing the importance of HWC in local food production, the Royal Government of Bhutan approved the development of the HWC management strategy in 2008, to address HWC for the Royal Government's 10th five-year plan: 2008–2013. Consequently, a national strategy, titled "Bhutan National Human–wildlife Conflicts Management Strategy" [56], the first of its kind for Bhutan, was developed to reduce HWC and ensure wildlife conservation.

The management strategy document states that HWC was "absent two decades ago" [56] (p. 7) in Bhutan. However, the strategy document also points out that the same farmers who tolerated HWC in the past, have come forward demanding action from the government. This purported behavioral change in farmers may have come about due to the establishment of protected areas that restricted farmers' access to forest resources [57], as over half (51.3%) of the country has now been declared protected areas, including national parks and biological corridors [58].

Literature from Bhutan [59–63] shows incidents of HWC are concentrated in settlements close to national parks, similar to the broader HWC literature [64,65]. Letro et al. [66] confirmed the large diversity of wild animals in the protected area network. According to Katel et al. [60] and Wang et al. [62], increased wildlife populations have increased threats to humans, livestock, and wild animals, creating conflicts among local people and park management officials, and impacting the food self-sufficiency goal of the country. While wildlife also exists outside of protected areas, proximity to a protected area would appear to increase the likelihood of HWC in rural areas of Bhutan [56,62].

Given this context, of the perceived increase in the incidence of HWC decreasing farmers' tolerance of wildlife damage, and limited agricultural land, the primary objective of this article is to explore the implications of HWC to Bhutan's aspiration to be food self-sufficient. We expect HWC is a concern for rural homesteads, with large proportions of the farm produce lost to wild animals [14,17,24], and other factors such as climate and environmental changes [28,29,49], due to the findings of previous studies. Therefore, in order to achieve the primary objective of the study, we asked, "what was the extent of crop damaged by wildlife in the last three years (2017–2019)?", to estimate the self-reported quantity of food lost to wild animals by households, and to identify the wild animal species most commonly perceived as problematic by farmers.

## 2. Methodology

The study adopted a mixed methods approach [67], involving a three-step data collection process. The study began with semi-structured interviews with key informants, followed by a questionnaire survey with household participants. Finally, another round of semi-structured interviews with household participants and key informants was conducted, as outlined in the following sub-sections. The objective of the first key informant interviews was to inform the development of the questionnaire, and the second key informant interviews were conducted to validate and explain the survey results.

### 2.1. Case Study Area

Bhutan is divided into three broad eco-floristic zones based on the forest types found there: alpine, temperate, and sub-tropical zones [68]. Each eco-floristic zone has characteristic fauna; however, some wild animals are found across all zones. For example, tiger

(*Panthera tigris*) is identified as one of the characteristic fauna of the sub-tropical zone, but tigers are also recorded in alpine forests [69]. Similarly, wild pigs (*Sus scrofa*) are reported as problematic animals throughout the country by farming communities [54,56,62,63].

According to the Forests and Nature Conservation Act of Bhutan, 1995 [70], all wild animals and plants listed in 'Schedule I' are fully protected, and other animals, "not listed in Schedule I, are also protected and may not be killed, injured, destroyed, captured, collected or otherwise taken (p. 11)". However, the act allows the killing of wild animals to defend against an attack on human life, livestock, and damage to crops, or according to hunting rules which may be issued by the relevant ministry [70]. Currently, there are no records of hunting rules issued by the government.

Administratively, Bhutan is divided into twenty districts, which are divided into small administrative units known as *Gewogs* (block). *Gewogs* are further divided into *Chiwogs* (sub-block). For this study, case study sites were selected after conducting a preliminary analysis of the 2019 house/farmland abandonment (known locally as *Gungtong*, see [71]) data. The data were obtained from the Department of Local Government, Royal Government of Bhutan. We first selected two districts, within which we chose two *Gewogs*. Finally, two *Chiwogs* within each study *Gewog* were selected for the study. While the district and *Gewogs* were selected based on the percentage of *Gungtong* households, *Chiwogs* were chosen randomly.

Two districts, shown in Figure 1, Trashiyangtse and Tsirang, were selected. Trashiyangtse has the highest percentage of *Gungtong* households at the district level in Bhutan, at 20.6% of the total households in 2019. Conversely, Tsirang district was selected because one of its *Gewogs*, Barshong *Gewog*, had the highest percentage of *Gungtong* households amongst all *Gewogs* in Bhutan, at 38.9%.

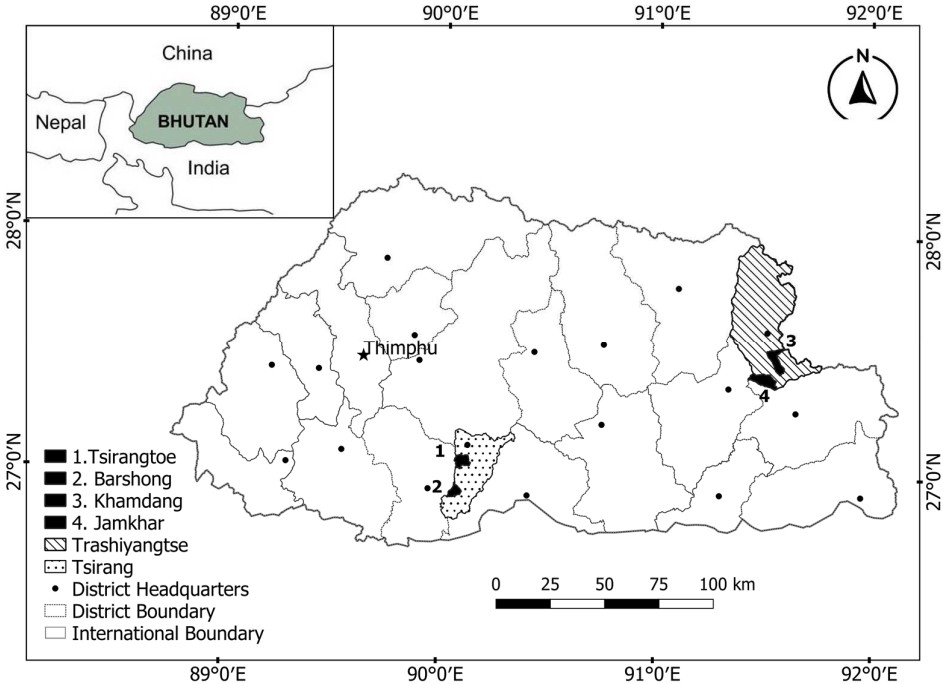

**Figure 1.** Map of Bhutan, showing study sites.

Trashiyangtse district is located in the country's eastern region, and is divided into eight *Gewogs*. The district is at an elevation of 500–5401 m.a.s.l., with an area of 1438.8 km$^2$. Most villages in the district are located along hill slopes, ranging from gentle to medium gradient [72]. The northern half of the district is home to Bomdeling Wildlife Sanctuary (BWS), which borders Wangchuck Centennial National Park (WCNP) to its west, see [73]. BWS is also connected to Phrumsengla National Park to its south, by an area designated as a biological corridor. According to NSB [74], the district's population in 2020 was 16,960,

and had 3581 households. The households in the district owned a total of 971, 4375, and 0.2 hectares of dry land, irrigated, and orchard lands, respectively, in 2020.

Most villages in the district are located within the temperate eco-floristic zones of the country, dominated by three broad forest types: fir, blue pine, and broadleaf mixed with conifer forest [68]. Residents of the district mostly grow paddy and maize for consumption, and potatoes and chilies for trade. Two *Gewogs*, Jamkhar and Khamdang, were selected for the study, as Jamkhar had the highest, and Khamdang had the lowest, *Gungtong* percentage in the *Gewog*, in 2019.

Tsirang district is divided into twelve *Gewogs* and is located in the country's central-western region. Unlike Trashiyangtse district, there is no designated protected area in Tsirang, nor is the district connected to any biological corridors, although there are two protected areas and biological corridors in the adjoining districts. Tsirang has an area of 639 km$^2$ and ranges from 500–1900 m.a.s.l., with most settlements along the gentle and medium hill slopes [75], in the sub-tropical eco-floristic zone, dominated by broad-leaf, chir pine, and tropical low-land forests [68]. As per the records maintained by NSB [74], the district had a population of 23,493 people in 2020, in 4254 households, and the district is thus more heavily populated than Trashiyangtse district, with almost 30% higher population in less than half the area. In total, the households owned 3142, 1659, and 402 hectares of dry land, irrigated, and orchard lands, respectively. Barshong and Tsirangtoe *Gewogs* were selected for the study. Barshong had the highest, and Tsirangtoe the lowest, *Gungtong* percentage in the *Gewog*, in 2019. Agriculture and livestock farming are the primary sources of livelihood for the majority of the population in the study *Gewogs*. Households in Tsirang district depend on vegetable and horticultural crops such as mandarin and cardamom for income, and grow paddy and maize on most of their lands.

*2.2. Data Collection and Analysis*

This manuscript is part of a broader study, exploring the links between HWC and migration, thus the broader set of questions asked to achieve the objective is available in the thesis, see [76]. Initial key informant interviews (n = 16) were undertaken face-to-face in October 2019. These initial interviews aided in developing the household survey, which was undertaken in selected *Chiwogs* of Trashiyangtse and Tsirang districts. Key informant participants included government and local government officials and household members. Government officials were selected based on their work duty, and local government officials included elected representatives of people in the *Gewog*. Household participants were those members from *Gungtong* (households who abandoned farmlands in the *Gewog* and migrated) and non-*Gungtong* (household members who were still living in the *Gewog* during the interview) households.

The lists of *Gungtong* and non-*Gungtong* households were obtained from the *Gewogs*. A simple random sampling was administered to the list of households, using the "=RAND" function in Microsoft Excel, to select the first *Gungtong* and non-*Gungtong* household participants. However, if the members of the first selected household were not available for interview, the next randomly selected household was approached. After the first interview, the rest of the households were approached, using the snowball technique [77]. We asked the existing research participants to identify our future participants, based on their acquaintances and their perceived knowledge of the subject, and gathered contact information such as our next participant's name and phone number.

Our second step of data collection was a questionnaire survey. The questionnaire survey was undertaken face-to-face in two selected *Chiwogs* in each study *Gewog*, during November and December 2019, resulting in 431 non-*Gungtong* responses [Trashiyangtse (n = 226) and Tsirang (n = 205)]. An effort was made to administer the survey to the head of each household in the selected *Chiwog* and, if unavailable, any present adult from the household. The questionnaire survey consisted of both open and closed-ended questions. To evaluate the type and extent of conflicts, all survey respondents who reported having experienced conflicts with wild animals were asked, "what was the extent of damage by

wildlife in the last three years (2017–2019) to your farm?" Within this question, respondents were asked to, (1) name wild animals that affected them, (2) state the kind and the extent of damage, such as the area of crops damaged or monetary value of crops destroyed, stored grains lost in kilograms, and finally, (3) state the frequency of these incidents. The reported crops and stored grains damaged by wild animals were converted to standard measurement units based on the "Standardization of measurement unit survey, Bhutan" [78], and the "Cost of production for field and horticulture crops in Bhutan" [79].

The third step of data collection involved another round of key informant interviews (n = 24), pursued virtually, after the survey (October 2019 to February 2021), to aid validation and explanation of the survey findings. They had to be conducted virtually, through Zoom, WhatsApp, and WeChat, as face-to-face interviews were not possible due to COVID-19. Interview participants were government officials, local government officials, journalists, and household members. Unlike the initial key informant interviews, for this step, household participants were those who demonstrated a substantial understanding or insight during the survey into the key issues of migration, *Gungtong*, and HWC. Both stages of key informant interviews were conducted until data saturation was reached.

Both interviews and surveys were conducted in the local language and translated into English. The interviews were transcribed. The qualitative analysis program NVIVO 12 Pro was used during the analysis, to assist with data organization during coding and identifying themes. We used IBM SPSS Statistics 28.0 to analyze the quantitative data.

## 3. Results

The vast majority of respondents from both study districts (Trashiyangtse = 98.7%, Tsirang = 92.2%) reported having experienced conflicts with wild animals, in the form of crop damage and livestock predation, within the last three years (2017–2019). Though households lost livestock to wild animals, conversations with respondents generally focused on crop damage, indicating the importance of crops to households in the study area. This article refers to cash crops as orchards, such as mandarin and cardamom.

Based on the respondents' self-reported crop damage by wild animals, households from Trashiyangtse suffered a significantly higher wild animal impact (crops lost ($p < 0.001$), and stored grains lost ($p = 0.004$)) compared to households from Tsirang (Table 1). We estimated a higher area of crop damage for households from Trashiyangtse (10.2%) compared to Tsirang (2.7%). It must be noted that the proportion of crop damage (to cultivated land) may have been much higher if fallow lands were removed from the equation.

**Table 1.** An independent sample t-test of crop damage, and stored farm produce and cash crops (i.e., orchards) lost to wild animals, between respondents from the Trashiyangtse and Tsirang districts.

| Type of Conflicts | | Levene's Test for Equality of Variances | | | | *t*-Test for Equality of Means | | | | | | |
|---|---|---|---|---|---|---|---|---|---|---|---|---|
| | | Mean | SD | F | Sig. | t | df | Sig. (2-Tailed) | Mean Difference | Std. Error Difference | 95% Confidence Interval of the Difference | |
| | | | | | | | | | | | Lower | Upper |
| Crops lost in area (ha) | Trashiyangtse | 0.4054 | 0.46 | 53.767 | <0.001 | 6.312 | 318.715 | <0.001 | 0.231 | 0.037 | 0.158 | 0.302 |
| | Tsirang | 0.1746 | 0.23 | | | | | | | | | |
| Stored farm produce lost in kg | Trashiyangtse | 284.72 | 360.10 | 10.140 | 0.002 | 2.924 | 86.707 | 0.004 | 154.169 | 52.720 | 49.374 | 258.963 |
| | Tsirang | 130.55 | 144.64 | | | | | | | | | |
| Orchards lost in values (Nu.) | Trashiyangtse | 4485.71 | 5098.91 | 0.842 | 0.367 | −0.979 | 20.780 | 0.339 | −2838.095 | 2899.033 | −8870.846 | 3194.656 |
| | Tsirang | 7323.81 | 9924.49 | | | | | | | | | |

### 3.1. Problematic Wild Animals

Four wild animals, wild pigs (*Sus scrofa*), monkeys (*Macaca assamensis*), porcupines (*Hystrix indica*), and barking deer (*Muntiacus muntjak*), were identified by most respondents as the worst crop raiders in both districts. Interactions with serrow (*Capricornis thar*), langur (*Semnopithecus entellus*), and mice (*Rattus* spp.) were less frequent (Table 2). Among the worst crop raiders, wild pigs were most often reported by respondents (Trashiyangtse = 94.7%, Tsirang = 84.8%) to have inflicted damage (highest reported average area of crop damage as

compared to other animals). Regarding damage to stored grains and orchards, respondents reported monkeys inflicted greater damage.

**Table 2.** Estimates of the crop damage inflicted by wild animals from 2017–2019 on respondents' households, from the two study districts. [N = total number of respondents; n = respondents who reported; SD = Standard deviation).

| | Trashiyangtse (N = 223) | | | | Tsirang (N = 190) | | | |
|---|---|---|---|---|---|---|---|---|
| | n | Mean | Sum | SD | n | Mean | Sum | SD |
| *Wild animals that destroyed field crops, by area (ha) \** | | | | | | | | |
| Wild pig (*Sus scrofa*) | 198 | 0.28 | 55.60 | 0.33 | 151 | 0.11 | 16.40 | 0.15 |
| Barking deer (*Muntiacus muntjak*) | 68 | 0.13 | 8.60 | 0.14 | 100 | 0.05 | 5.00 | 0.06 |
| Monkey (*Macaca assamensis*) | 60 | 0.20 | 11.80 | 0.25 | 82 | 0.07 | 5.38 | 0.10 |
| Porcupine (*Hystrix indica*) | 53 | 0.12 | 6.60 | 0.18 | 34 | 0.05 | 1.72 | 0.07 |
| Sambar (*Rusa unicolor*) | 7 | 0.10 | 0.70 | 0.07 | | | | |
| Birds | 5 | 0.17 | 0.90 | 0.18 | 12 | 0.07 | 0.83 | 0.14 |
| Squirrel (*Petaurista* spp.) | 1 | 0.25 | 0.25 | | 10 | 0.05 | 0.45 | 0.04 |
| Himalayan black bear (*Ursus thibettanus*) | 1 | 0.14 | 0.14 | | 10 | 0.04 | 0.39 | 0.03 |
| Serrow (*Capricornis thar*) | 1 | 0.05 | 0.12 | | 3 | 0.04 | 0.11 | 0.02 |
| Mice (*Rattus* spp.) | | | | | 9 | 0.05 | 0.41 | 0.03 |
| Langur (*Semnopithecus entellus*) | | | | | 2 | 0.02 | 0.04 | 0.03 |
| Total | | | 84.71 | | | | 30.73 | |
| *Wild animals that destroyed stored grains, by weight (kg)* | | | | | | | | |
| Monkey (*Macaca assamensis*) | 55 | 252.69 | 13,897 | 351.61 | 29 | 90.31 | 2617.63 | 121.01 |
| Birds | 14 | 186.39 | 2609 | 207.98 | 1 | 40.00 | 40.00 | |
| Squirrel (*Petaurista* spp.) | 4 | 191.25 | 766 | 135.18 | 2 | 0.09 | 0.18 | 0.10 |
| Himalayan black bear (*Ursus thibettanus*) | 1 | 96.00 | 96 | | 6 | 246.67 | 1480.00 | 156.93 |
| Langur (*Semnopithecus entellus*) | | | | | 1 | 40.00 | 40.00 | |
| Total | | | 17,368 | | | | 4177.8 | |
| *Wild animals that destroyed cash crops, by value (USD) \*\** | | | | | | | | |
| Monkey (*Macaca assamensis*) | 5 | 65.40 | 327.10 | 87.40 | 18 | 110.40 | 1987.90 | 153.70 |
| Sambar (*Rusa unicolor*) | 3 | 28.60 | 85.70 | 19.60 | | | | |
| Barking deer (*Muntiacus muntjak*) | 1 | 35.70 | 35.70 | | | | | |
| Wild pig (*Sus scrofa*) | | | | | 4 | 34.50 | 139.30 | 21.70 |
| Langur (*Semnopithecus entellus*) | | | | | 2 | 35.00 | 70.00 | 29.30 |
| Total | | | 448.5 | | | | 2197.2 | |

\* Losses were reported in acres. \*\* Losses were reported in Ngultrum (Nu.), and were converted to USD. 1USD ~ Nu. 70.00.

Most respondents reported having conflicts with wild pigs daily (Trashiyangtse = 77.3%, Tsirang = 67.5%). Based on the reported crop damage, 65.6% and 53.4% of the area of crop damage was inflicted by wild pigs, followed by monkeys (Trashiyangtse = 13.9%, Tsirang = 15.7%). However, monkeys were identified as responsible for about 80% and 62.7% of the total stored grains damaged by wild animals in Trashiyangtse and Tsirang, respectively.

Therefore, based on what was reported by respondents, wild pigs are the most problematic species in the study area, followed by monkeys. Most respondents perceive that the population of wild pigs has increased, and claim that, " . . . sometimes I have seen more than 30 wild pigs in a group . . . I wish we are allowed to hunt them . . . " Another said, " . . . five years ago, wild pigs damaged my whole maize farm . . . and my family had to depend on monies sent by our children . . . " Such statements indicate the desire of households to hunt wild pigs, and the dependence of households on remittances when their crops are lost to wild animals.

### 3.2. Crops Lost to Wild Animals

Wild animals deprive households of their food as well as their income source. The vast majority of respondents (Trashiyangtse = 91.9%, Tsirang = 83.2%) reported having lost maize, with the next largest loss being paddy (Trashiyangtse = 31.4%, Tsirang = 35.3%) (Table 3). These two crops are widely cultivated by households in the study area and form the main staple diet of Bhutanese people. In addition, 22.4% and 18.4% of respondents from Trashiyangtse reported that wild animals raided potatoes and vegetables (primarily chilies), respectively, which are their primary cash income source. Similarly, 33.2% and 11.1% of Tsirang respondents reported that wild animals damaged vegetables (mostly beans) and orchards, respectively.

**Table 3.** Estimates of the type of damage inflicted by wild animals, from 2017–2019, on respondents' households, from the two study districts. [N = total number of respondents; n = respondents who reported; SD = Standard deviation).

| | Trashiyangtse (N = 223) | | | | Tsirang (N = 190) | | | |
|---|---|---|---|---|---|---|---|---|
| | n | Mean | Sum | SD | n | Mean | Sum | SD |
| Area of crops lost to wild animals (ha) * | | | | | | | | |
| Maize (*Zea mays*) | 205 | 0.28 | 57.38 | 0.33 | 158 | 0.11 | 17.75 | 0.19 |
| Paddy (*Oryza sativa*) | 70 | 0.17 | 11.71 | 0.2 | 67 | 0.09 | 6.16 | 0.10 |
| Potato (*Solanum tuberosum*) | 50 | 0.13 | 6.67 | 0.15 | 5 | 0.06 | 0.30 | 0.04 |
| Vegetable | 41 | 0.09 | 3.61 | 0.12 | 63 | 0.03 | 2.02 | 0.04 |
| Groundnut (*Arachis hypogaea*) | 24 | 0.12 | 2.76 | 0.14 | | | | |
| Soybean (*Glycine max*) | 13 | 0.09 | 1.17 | 0.16 | 4 | 0.03 | 0.13 | 0.03 |
| Pulses (*Lens* sp.) | 6 | 0.17 | 1.00 | 0.14 | 52 | 0.05 | 2.48 | 0.07 |
| Other cereals | 5 | 0.08 | 0.42 | 0.09 | 39 | 0.04 | 1.42 | 0.04 |
| Tapioca (*Manihot esculenta*) | | | | | 17 | 0.03 | 0.47 | 0.03 |
| Total | | | 84.72 | | | | 30.73 | |
| Weight of stored farm produce lost to wild animals (kg) | | | | | | | | |
| Maize (*Zea mays*) | 59 | 259.70 | 15,324 | 341.10 | 30 | 115.80 | 3472 | 134.90 |
| Vegetable | 6 | 120.00 | 720 | 130.20 | | | | |
| Paddy (*Oryza sativa*) | 3 | 209.70 | 629 | 234.20 | 6 | 107.00 | 642 | 59.40 |
| Potato (*Solanum tuberosum*) | 3 | 213.30 | 640 | 142.90 | 2 | 11.50 | 23 | 10.60 |
| Soybean (*Glycine max*) | 1 | 55.00 | 55 | | | | | |
| Pulses (*Lens* sp.) | | | | | 1 | 40.00 | 40 | |
| Total | | | 17,368 | | | | 4177 | |
| Value of cash crops lost to wild animals (USD) ** | | | | | | | | |
| Orchards | 7 | 64.1 | 448.5 | 72.8 | 21 | 104.6 | 2197.2 | 141.8 |
| Total | | | 448.5 | | | | 2197.2 | |

* Losses were reported in acres. ** Losses were reported in *Ngultrum* (Nu.), and were converted to USD. 1USD ~ Nu. 70.00.

As a guideline to estimate the food expenditure lost, we referred to the Living Standards Survey Report of Bhutan [80]. The report states that the monthly mean household food expenditure in rural Bhutan was estimated at Nu. 12,606 (USD 180.00) in 2017. Based on the respondents' data, we estimated the mean value of food expenditure loss as USD 189.6 (USD 1.1 to USD 1480.7, median = USD 95.5), for respondent households from Trashiyangtse, and USD 80.00 (USD 0.63 to USD 750.3, median = USD 46.3), for Tsirang respondents' households.

On average, respondent households from Trashiyangtse and Tsirang lost over half a month to more than a month's worth of household food expenditure to wild animals annually, from 2017–2019. As shown in Table 4, 9.1% of respondents' households from Trashiyangtse lost 3–6 months' worth of food expenditure to wild animals annually, from

2017–2019, and only 1.2% of respondents' households from Tsirang reported having lost the same amount (Table 4).

**Table 4.** Percentage of respondents from two study districts, showing the annual crop loss from 2017–2019 regarding household food expenditure.

| Household Food Expenditure Lost | Percentage of Respondents | |
|---|---|---|
| | Trashiyangtse [N = 223] | Tsirang [N = 190] |
| <1 month | 68.4% | 90.4% |
| 1 to 3 months | 22.0% | 8.4% |
| 3 to 6 months | 9.1% | 1.2% |
| >6 months | 0.5% | |

A respondent from Trashiyangtse claimed to have lost over eight months' worth of food annually, from 2017–2019, to wild animals, stating, " . . . a family of wild pigs made my farm their home . . . don't want to work anymore on the farm . . . " This statement from the respondent suggests the role HWC plays in the decision of farmers to discontinue farming, which has consequences for local food production.

*3.3. Implications for Food Self-Sufficiency*

The perceived increase in the incidence of HWC events drives farmers to abandon their farmlands and explore other livelihood options. For example, a respondent said, " . . . those households with farmlands at the edge of the settlement leave their land fallow as they cannot bear the damages caused by wild animals . . . " Another said, " . . . some households are left with no adequate harvest, which becomes a concern for households' survival . . . " One of the consequences of not producing adequate food is the migration of the working-age population, which translates to a farm labor shortage and, ultimately, an increase in fallow lands and reduced farm production.

The perceived increase in fallow lands is perceived to bring wild animals closer to settlements. According to most respondents, reducing farming is reported to impact other farmers as, " . . . fallow lands bring wild animals closer to settlement . . . " (Trashiyangtse = 56.4%, Tsirang = 45.9%). For example, a respondent from Trashiyangtse said, " . . . my neighbor is worried after most of his crops were destroyed by wild animals last year and even this year. So, they are planning to discontinue farming . . . I'm worried that once his fields are left fallow, wild animals will destroy my crops . . . " This means, with households leaving increased areas of land fallow, HWC events are perceived to increase. This opinion was consistent between both survey and interview respondents.

With the perceived increase in wild animals coming closer to settlements, households spend considerable time guarding their crops, in addition to fencing, making scarecrows, making fire, and even placing stuffed tiger dolls in the field. For 82% of respondents, who reported having experienced HWC from 2017–2019 (n = 413), guarding was one of the important measures practiced to protect their crops from wild animals. One respondent said, " . . . today is the second night I slept in my house . . . I spent eight months sleeping in my makeshift hut guarding my crops . . . though we fenced our field, we have to guard day and night as soon as we sow the seeds . . . "

However, despite investing many resources into protecting crops from wild animals, households still lose much of their crop. The continued loss of crops to wild animals potentially drives households to discontinue farming. For example, a household interview participant in Tsirang explains that, " . . . it is best to work somewhere as a laborer as we do not have to worry about wild animals . . . if we work for two days, we can earn enough to buy a bag of rice which can last for about 15 to 20 days. If we work for a day, we can buy 5 litres of cooking oil . . . " Another interview respondent was very vocal and said, " . . . we have plenty of wild animals, and India produces plenty of rice and oils which are readily available in the market . . . " .

Taken together, household respondents perceive that HWC and fallow lands form a positive feedback loop. First, crop damage by wild animals contributes to a household's decision to discontinue farming, and second, HWC incidents are perceived to increase with fallow farmlands. A government official agrees with these narratives, and expresses that, " ... our ancestors grew their food, but we are now importing more food ... wild animals are to be blamed in addition to farm labor shortages ... " Therefore, households leaving more lands fallow is perceived to increase the incidence of HWC events, affecting household food self-sufficiency.

## 4. Discussion

The perceived implications of an increase in HWC events influence households' decisions about whether to continue farming or to explore other livelihood options. This is because farm production costs, due to guarding and crops lost to wild animals, can be so high for some households that it is easier to discontinue farming. This is consistent with reports from China [27] and Nepal [22], which described some croplands being left fallow due to severe crop damage by wild animals.

Fallow lands are already a concern for Bhutan. According to the Renewable Natural Resources (RNR) census of Bhutan, 26.4% of the total agricultural landholdings in the country were left fallow at the time of the census in 2019 [55]. The land use and land cover data in 2010, reported that agricultural land covered 3.13% of the country's total geographic area [81]. By 2016, only 2.75% of the country was under cultivation [42]. Such a reduction in cultivated land is a concern for an already land-scarce and import-dependent country like Bhutan.

However, HWC is not the only factor leading to the increase in fallow lands in Bhutan. The RNR census of Bhutan found the shortages of irrigation water and labor as the top two most important reasons for lands being left fallow, followed by HWC [55]. Similarly, other studies have identified increased vulnerability to natural hazards [30], due to climate change [82], coupled with limited access to markets and poor economic returns [83], as some of the factors that discourage farming. Furthermore, in mountainous countries, steep slopes and less fertile soils [32], the distance of farmland from the settlement location, and increased labor costs [83,84] are some factors contributing to farmland abandonment.

We present a conceptual framework linking the increase in food imports with the perceived increase in HWC events, in Figure 2. Our framework shows that farmers will discontinue farming with the perceived rise in HWC events. This is consistent with several studies describing a relationship between the increase in fallow lands and HWC events [22,23,35]. The discontinuation or reduction in farming practices leads some households to explore other livelihood opportunities, and then migrate, reducing household farm labor and thereby increasing fallow land [85]. Li and Li [21] and Yan et al. [14] reported a similar process in China, of abandoning farmlands followed by leaving the location. Furthermore, literature from different parts of the world revealed that migration negatively impacts local farming and food self-sufficiency through labor shortages [82,84–86].

Further, the discontinuation of farming by households increases fallow land, which contributes to the increase in forest regeneration, expanding the wildlife's habitats [5,21]. The consequences of expanding the wildlife's habitats have a direct link to an increase in HWC events [23,38], which can lead to increased food shortages and poverty among marginal and small farm households in the villages, as reported by Khanal and Watanabe [87] in a mountain community of Nepal. Thus, as shown by our framework (Figure 2), and consistent with several other studies [5,21–23], the increase in HWC events potentially leads to an increase in fallow lands, which in turn increases HWC incidents. This feedback loop ultimately leads to reduced food production, which translates to an increased dependency of subsistence homesteads on purchased food [19], thereby increasing food imports. The increased dependency on purchased food could mean farmers struggling to buy food, with not enough money, as reported by Leahy et al. [40] in Zambia.

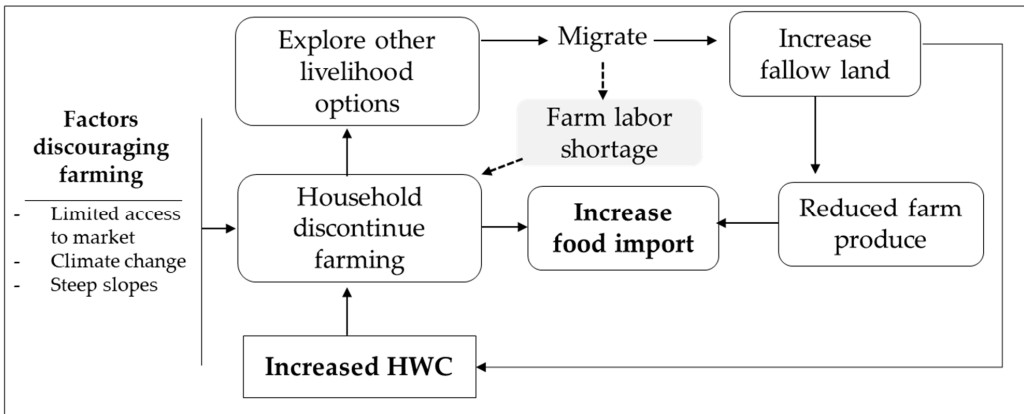

**Figure 2.** A conceptual framework showing the relationship between HWC and food imports.

Regarding the estimates of crop loss to wild animals, we could not verify the claims of loss reported by respondents, as there were no official records to triangulate. Despite the self-reported crop loss data, our results suggest that households from Trashiyangtse experienced a higher incidence of HWC events than Tsirang households. A possible explanation for Trashiyangtse households experiencing a higher incidence of HWC could be the proximity of Trashiyangtse households to a protected area (Bomdeling Wildlife Sanctuary, in northern Trashiyangtse). Further, Trashiyangtse district is connected to two national parks: Wangchuk Centennial National Park to its west, and Phrumsingla National Park to its south, through a biological corridor, which provides a 'safe' passage for wild animals to migrate, as reported by Letro et al. [66].

The higher value of crop loss by households from Trashiyangtse is consistent with the reports elsewhere, which state that the incidence of HWC is increasing worldwide, especially in and around protected areas [88]. Further, studies from Bhutan confirmed the presence of a large diversity of carnivores and herbivores within the country's protected area network [66]. In addition, a study conducted in the four districts of Bhutan by Yeshey et al. [89], found that economic losses through livestock and crop loss to wild animals were significantly higher for those residing inside the protected areas. This becomes a critical issue for Bhutan and elsewhere, as conservation activities are prioritized, but more than half of the population are subsistence farmers.

Conversely, Tsirang district has no designated protected areas, nor is the district connected to any biological corridors. Perhaps, Trashiyangtse residents, being closer to the protected area network, are experiencing a higher incidence of HWC events, resulting in more fallow farmlands than in Tsirang. According to the agriculture census report, 45.4% of farmlands were left fallow in 2019 in Trashiyangtse, compared to 14.0% left fallow in the Tsirang district [55]. However, a thorough study is required to understand these differences in the HWC experience between the respondents from these two districts, though there also exists evidence from parts of Africa and India of HWC being more of a problem for households living near protected areas [90,91].

All wild animals identified as problematic species by our respondents are listed as *least concern* in the IUCN red list, except for the Himalayan black bear and sambar deer, which are listed as vulnerable. Similarly, the Forest and Nature Conservation Act of Bhutan [70] categorized the Himalayan black bear and serrow, as 'Schedule I', which means totally protected. This means all other wild animals identified as problematic animals are not totally protected in Bhutan, however the act does not allow hunting, except upon the issuance of hunting rules by the government, the record of which does not exist.

This study identified wild pigs as causing greater crop damage than other wild animals. This is consistent with several other studies in the Indo-Himalayan region. Pandey et al. [92] (p. 107) asserted wild pigs as being the "primary crop raider and driver of HWC", in their study site in Nepal. Perhaps, recognizing the severity of crop damage caused by wild pigs, the Government of Bhutan developed a strategy document [56] identifying

regulated hunting as one of the strategies to reduce the wild pig population. However, this strategy has never been adopted. Since other wild animals were not reported to be as destructive as wild pigs by our respondents, the following discussion focuses on wild pigs.

Though fewer respondents from our study sites mentioned hunting wild pigs as a solution to the problem, there seems to be increasing support, with some expressing their wish for the government to legalize hunting. Perhaps, the Buddhist ethics of tolerance and living in harmony with nature, a widespread perspective of Bhutan's rural populace, is changing with the increasing incidence of HWC. A study from other parts of Bhutan reported that most of their study respondents "expressed a strong desire to exterminate problem wildlife" [62] (p. 153).

There exists evidence of reduced HWC after relaxing some conservation policies. Some countries have adopted culling as a control method for the wild pig population. For example, the use of lethal methods for wild pig control was legalized in 2013 in Brazil [93]. A population model by Croft et al. [94] suggests a combination of fertility control and culling, as the most cost-effective method of reducing the wild pig population. Similarly, but not for wild pigs, an analysis of over 40 years of data on the human–bear conflict in the United States, by Garshelis [95], found that complaints about the conflict remained relatively low after hunting bears was legalized.

Therefore, in congruence with other studies, we call for controlled hunting of wild pigs, to reduce HWC, after carefully assessing their population status. Further, culling of selected animals could potentially generate resources for conservation of all species [96]. There are reports of wild pigs being a threat to conservation of some endangered floral species, by damaging regeneration, especially in wetland areas [97]. However, a detailed study needs to be pursued, to understand the general perception of people regarding the control measures, as König et al. [4] suggest people's perceptions as being central to achieving coexistence. If culling is to be instituted, it becomes critical to streamline various elements, such as preventing excessive culling, and educating farmers [98].

It is important to note that, our article concerns the negative outcomes of human and wild animal interactions on farming communities. We acknowledge that there is evidence of negative outcomes of HWC on wild animals [10,11], through retaliation and habitat loss [13]. Thus, there is an increased risk of developing negative attitudes to conservation [10,12,62] if the incidence of HWC remains high, and this could perhaps be detrimental to endangered species.

However, Bhutan is mainly Buddhist—the philosophies and beliefs associated with this religion discourage killing animals. Thus, proposing any interventions to control the population of problem wild animals must be carefully crafted, before the rural landscape of Bhutan stops producing food. Further, some studies state that Bhutan's conservation policies are "disproportionately skewed towards the conservation of nature" [51] (p. 198), without addressing human aspects [99]. Therefore, it remains crucial for Bhutan to focus on the delicate balance between conservation and farming, with farmers struggling to support their livelihood. Taken together, if the underutilization and abandonment of farmland continues, achieving food self-sufficiency is an implausible dream for Bhutan.

## 5. Conclusions

The primary objective of our article was to explore the implications of HWC on Bhutan's aspiration to be food self-sufficient, through quantifying the self-reported crops and stored gains lost to wild animals. Our study reveals the impact of wild animal crop damage on farming communities' food self-sufficiency, which in turn is linked to the national food self-sufficiency goal. Though there are other factors, such as a shortage of irrigation water and farm labor, the increase in HWC events will undoubtedly impact food self-sufficiency, as it has a direct link with households discontinuing farming, and increasing fallow lands.

Among the ten wild animals, and some species of birds, identified as problematic animals by our research participants, wild pigs were most widely reported to cause damage to crops, both in the frequency of incidents and the quantity of crops damaged. Perhaps

wild pigs are the 'greatest enemy' of farmers, and one of the many factors discouraging farming.

Therefore, if no reliable permanent measures are adopted by the government, other than temporary responses such as providing electric fences and compensation [52] for the loss of crops and livestock, it is likely that Bhutan will increasingly import more food, and the country's rural landscape may be abandoned.

Further, with those households near protected areas losing much of their crop to wild animals, people's support for conservation may be diminished, which could be detrimental to endangered species through retaliation. In addition, it is only a matter of time before major social crises emerge, such as a conflict between households and conservationists, and the abandonment of the rural landscape, seriously impacting local food production.

**Author Contributions:** S.W.: conceptualization, methodology, formal analysis, investigation, writing—original draft, writing—review and editing. J.B.: validation, supervision, writing—review and editing. R.T.: validation, supervision, writing—review and editing, M.F.: validation, supervision, writing—review and editing. All authors have read and agreed to the published version of the manuscript.

**Funding:** This research was funded by the Australian Government Research Training Program and Gulbali Institute Bridging funding.

**Institutional Review Board Statement:** The study was conducted in accordance with the Human Research Ethics, approved (Protocol number: H19242) on 2 September 2019 by the Human Research Ethics Committee of Charles Sturt University.

**Informed Consent Statement:** Informed consent was obtained from all subjects involved in the study. However, written informed consent will not be available for publication to protect the privacy of the subjects.

**Data Availability Statement:** To protect the privacy of the subjects, data will not be available.

**Acknowledgments:** The authors wish to thank the Australian Government Research Training Program Scholarship and Gulbali Institute Bridging funding, for supporting the first author. We are grateful to all the participants involved in the household survey and interviews. Our heartfelt gratitude to the Ugyen Wangchuck Institute for Conservation and Environment Research (UWICER) in Bhutan for all logistical support, and to all our research assistants involved in data collection. Finally, we thank Kuenzang Dorji, researcher at UWICER for helping with the map works.

**Conflicts of Interest:** The authors declare no conflict of interest.

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
