# Peer review of "Exploring Human–Wildlife Conflict and Implications for Food Self-Sufficiency in Bhutan"

_sustainability, doi:10.3390/su15054175_

Round 1
Reviewer 1 Report
The field of human-wildlife conflict began in agricultural communities inconvenienced by wild animals: “the elephants raided my crops” or “the wolves ate my sheep.” This view is backed by an anthropocentric view of the world, one that only considers the preferences of humans (and in extreme cases, particular classes of humans) worthy of consideration. That is the approach taken by Wangchuk et al. They interviewed people in the agricultural sectors of two districts in Bhutan and found that most of them self-reported crop damage. They further identified the main causes of the damage to be wild pigs, barking deer, and monkeys. They cite a small number of people stating that landowners on the edges of settlements have so much damage that they “abandon their farmlands and explore other livelihood options” (l 288) and worry that this will lead to “the migration of the working-age population, which translates to a farm labour shortage and, ultimately, an increase in fallow lands and reduced farm production.” They imply that the source of the damaging wildlife is protected areas (although one of the two regions studied “has no designated protected areas, nor is the district connected to any biological corridors”, l 390-391) and that conservation measures preclude control of marauding animals, and “call for controlled hunting of wild pigs to reduce HWC after carefully assessing its population status” (l 421-422).
The authors have done a lot of work and represent the views of the agricultural sector well. However, there are some major issues here, not least of which is that the information reported by respondents is taken as gospel. There have been many studies (though perhaps not from Bhutan) showing that farmer perceptions of animal damage tend to be substantially elevated, relative to objective measurements. There is clear unconcern about the non-human components of the equation shown, for example, when the authors neglect to report the scientific names of the offending animals. Are the “wild pigs” a native species or feral agricultural animals? Are the “monkeys” an abundant species or a highly endangered one? From a sustainability/conservation perspective, those are important distinctions. Alternative approaches to conflict reduction are not really explored and important questions are not addressed. Are these animals really associated with protected areas? And if so, what economic benefits do those areas confer on their neighbors that might offset the damage? Is the movement of people from rural areas truly caused by the conflict, or could it be associated with the global phenomenon of agrarian decline and urban growth? Finally, the authors are also quite unfamiliar with the less-utilitarian literature that has emerged in the field in recent decades. For example, the recent volume edited by Angelici and Rossi (2020) is never cited, nor is its predecessor (2016).
Reviewer 2 Report
Dear Authors,
The manuscript is interesting and could be of a potential reader’s interest. The authors raise a very important issue of the conflict between wildlife and farmers. Unfortunately, the manuscript is quite chaotic and needs substantial work to clarify the content. My main concerns include luck of stated clearly research questions and hypotheses, messy methodology and results sections. Some fragments of the discussion and conclusions are not convincing and skewed and in contradiction to statements in the introduction (see lines 33-36).
General remarks
- All tables and graphs need more details in the captions/titles – they should be self-explanatory;
- Tables should be organised in the same manner using the same template of lines, fonts and symbols
- Provide units in parathesis e.g. should be [ha] instead of hectares (see tab 1)
- Explain all symbols used in the tables and use same notation (e.g. n vs N, P vs Sig) in the table caption not in the main body of the table
- Use same number of decimal places in the columns;
- Use SD and SE for standard deviation and standard error respectively
- Provide Latin names for the animal species in the text,
- There are repetitions of the same arguments both in the introduction and discussion, e.g. lines 50-51 vs 370-371, and 101-102 vs 337-339.
Introduction
It is in my opinion the best chapter of the manuscript. I would fused general part with a detailed part on conflicts in Buthan as a one part and clearly state the aims and research questions at the end.
Methodology
Description of the study area should provide general information about the study sites such as topography of both study sites (authors later discuss the difficulties of farming due to steepness of the mountainous areas), vegetation and main wild animals. Are wild animals the same in both study sites? What are population number estimates, status of protection etc?
Also description of administrative units is quite confusing. Is it important for the reader to know these names, especially that later the authors use only two main names? If still needed I suggest to move to supplementary section and leave in the main text only information of the total number of interviews.
The map does not explain in the legend red and green colours.
Data collection needs clarification. So called key informants – how they were chosen? What was the initial/target number of interviewees, what were criteria for the choice of the respondents? All these should be clearly stated. Also in 3rd step there is no information about the number of the respondents.
L. 209 – explain “substantial understanding of the subject”
For each “step” it should be explained why it was used. What was the scenario of the questionnaire, how many questions in total, preferable such a template should be attached in the supplementary material.
Line 211- 214. What exactly did the authors test – here you can provide hypotheses and the tests which were used not only information on the program.
Results
Should correspond with the methods
If the authors used three steps, for each should provide also the results
There are results for the “household respondents” and no for key informants
I suggest to provide information what is the formal status of the specific animal species mentioned in the results, i.e., are they protected, game species (if hunting is not allowed at all it should be mentioned in the description of the study area), are they threated (see IUCN red lists) etc.
I suggest reorganisation of the tables (see above in the general remarks)
What n, sum stand for?
Provide range (min -max) (tab 2, 3)
All longer explanations should be included in the table caption and not in the main body of the table (see tab 3)
What is cash crops?
The research describes data collected only in 2019, in table 4 there is information about crop loss between 2017- 2019. This must be mentioned in the methodology as well. As later the authors discuss increase of HWC over years it should be shown clearly in the results what happened in each year.
3.3 - I suggest categorisation of the respondents answers and shortage of the citated responses, also all additional authors comments should be moved to discussion part (see e.g. lines 305-307)
Discussion
It is mostly focused on wild pigs, how about other animals mentioned in the results?
Lines 403- 404 – explain why this strategy was not adopted
Lines 408 – 409 and 423 – 424 – repetition
Line 439 – provide references
Reviewer 3 Report
It is a very good work on the livestock producer-wildlife conflict, but it requires certain adjustments that we refer to below.
The study is confined to a Case Study, but it is not conceptually correct. It's actually a local observational study. The survey is a source of obtaining information. The implemented methodology seems correct, only that it would be necessary to explain in a little more detail how the ´snowball´ application was implemented in practice. This methodology should be supported bibliographically.
When presenting the problem to be explored in this region of Bhutan, the hypothetical information that is expected to be obtained should be made explicit, that is, what results are expected from the survey. Then, was the data of the respondents contrasted to ensure the veracity of what was answered? Clarify this please.
Finally, the Conclusion must be improved, here the objective of the work must be strictly answered and that is not clear in this section.
Reviewer 4 Report
This paper reports “Exploring human-wildlife conflict and implications for food self-sufficiency in Bhutan”. In general, the manuscript encompasses important issues regarding the human-wildlife conflict. Although the data were not previously available for this study site and the paper is valuable, it suffers from minor flaws. There are many comments concerning clarifications that are needed in the writing with special emphasis on the method and results sections and the addition of some statistical tests that I think would improve the strength of the manuscript. There are many discursive and self-contradictory sentences in the literature review and method sections without logical sequencing, lending to distractions and deviations from the key messages that the authors are trying to convey. Some of the comments concerning clarifications that are needed in the writing and that I think would improve the quality and strength of the manuscript are given in detail in the manuscript itself. The reference section is the very good one, written with better consistency. But it needs further polishing and checking if it is in line with the Sustainability journal style.

Round 2
Reviewer 1 Report
This revision is definitely improved, and I want to commend the authors for their effort. However, the underlying issue has not been addressed to my satisfaction and has to do with the very anthropocentric views expressed. This remains more appropriate for a local or agricultural journal than to an international journal focused on more holistic sustainability.
Reviewer 2 Report
Dear Authors,
The manuscript is improved, and the Authors provided substantial explanation and addressed to comments in the previous review. In my opinion the manuscripts could be accepted for publication.
Author Response
We are grateful to the reviewer for the detailed comments in our original version of the manuscript. We believe that the reviewer’s comment greatly improved the quality and structure of our manuscript. Thank you for recommending our article for publication.
Reviewer 3 Report
In full agreement with the review carried out, however, I think that a hypothesis or expected results should be made explicit, even when it is clear that it is an observational study.
Author Response
We thank the reviewer for all the comments on our original manuscript, which substantially improved the quality of our manuscript. In addition, as suggested by the reviewer, we made our hypothesis or expected study results explicit in this revision.
Line 136-138:
“We expect HWC is a concern for rural homesteads with large proportions of the farm produce lost to wild animals [14,17,24] and other factors such as climate and environmental changes [28,29,49] due to the findings of these previous studies.”